# Microscopic Methods for Identification of Sulfate-Reducing Bacteria from Various Habitats

**DOI:** 10.3390/ijms22084007

**Published:** 2021-04-13

**Authors:** Ivan Kushkevych, Blanka Hýžová, Monika Vítězová, Simon K.-M. R. Rittmann

**Affiliations:** 1Department of Experimental Biology, Faculty of Science, Masaryk University, 62500 Brno, Czech Republic; 448583@mail.muni.cz (B.H.); vitezova@sci.muni.cz (M.V.); 2Archaea Physiology & Biotechnology Group, Department of Functional and Evolutionary Ecology, Universität Wien, 1090 Wien, Austria

**Keywords:** microscopy, fluorescence microscopy, FISH, DAPI, *Desulfovibrio*, anaerobic microorganisms, habitats, SRB, SRP, SRM, sulfate reduction, identification, gut microbiota, IBD

## Abstract

This paper is devoted to microscopic methods for the identification of sulfate-reducing bacteria (SRB). In this context, it describes various habitats, morphology and techniques used for the detection and identification of this very heterogeneous group of anaerobic microorganisms. SRB are present in almost every habitat on Earth, including freshwater and marine water, soils, sediments or animals. In the oil, water and gas industries, they can cause considerable economic losses due to their hydrogen sulfide production; in periodontal lesions and the colon of humans, they can cause health complications. Although the role of these bacteria in inflammatory bowel diseases is not entirely known yet, their presence is increased in patients and produced hydrogen sulfide has a cytotoxic effect. For these reasons, methods for the detection of these microorganisms were described. Apart from selected molecular techniques, including metagenomics, fluorescence microscopy was one of the applied methods. Especially fluorescence in situ hybridization (FISH) in various modifications was described. This method enables visual identification of SRB, determining their abundance and spatial distribution in environmental biofilms and gut samples.

## 1. Introduction

Sulfate-reducing microorganisms (SRM) are a diverse group of anaerobic microorganisms, which are widely present in nature and play an indispensable role in the sulfur and carbon cycle on Earth [1]. This group comprises prokaryotes from the domains *Bacteria* and *Archaea*, encompassing over 220 species from 60 different genera [2,3,4]. Members of this exceptional physiological group differ from each other in their nutritional requirements and morphology; however, all its representatives use sulfate (SO_4_^2−^) or other oxidized sulfur compounds as a terminal electron acceptor in the oxidation of organic substances [2]. As this review deals mainly with the representatives of the domain *Bacteria*, the term sulfate-reducing bacteria (SRB) will be mainly used in the text.

SRB can use a wide range of substances as electron donors. For example, molecular hydrogen (H_2_) and various organic compounds (lactate, acetate, pyruvate, malate, alcohols such as ethanol, propanol or butanol and others) can serve as electron donors in anaerobic respiration [2,5]. Some SRB can also use nitrates as the final electron acceptor, for example, the representatives of the genera *Desulfovibrio* or *Desulfobacterium* [6,7,8,9]. Organic substrates can be oxidized by various species either incompletely to acetate (e.g., by the genus *Desulfovibrio*) or ultimately to carbon dioxide (e.g., by the genus *Desulfomicrobium*) [10]. The process is referred to as dissimilatory sulfate reduction or sulfate respiration. In this process, a small amount of reduced sulfur is assimilated, but most of it is released into the environment in the form of a sulfide ion, but usually as hydrogen sulfide (Figure 1). No other microorganisms than SRB are known to be capable of this form of respiration [11].

Intensive SRB studies started in 1895 when M. W. Beijerinck discovered an interesting biological activity in a newly isolated species, “*Spirillum desulfuricans*” (later *Desulfovibrio desulfuricans*). Indeed, it was the dissimilatory sulfate reduction activity that he discovered. Many other scientists went on to study SRB and new cultivation techniques of anaerobes, and the development of molecular techniques led to the characterization of many new taxa [2]. Recent studies are aiming to reveal the possible usage of SRB in biotechnology, too, where SRB can be used, for example, for bioremediation of toxic compounds in the environment [12,13,14,15]. Other studies investigate SRB’s capability to cause microbial corrosion and oil acidification, which is causing considerable economic losses. It is worth mentioning that SRB inhabit very diverse habitats. These organisms have successfully adapted to almost any ecosystem on Earth, from waters to soils and animal guts, humans included. SRB were detected in both the human intestines and oral cavity of patients with periodontitis [3]. Therefore, another important field of study considering SRB is their potential role in inflammatory bowel diseases or periodontitis, as SRB produce hydrogen sulfide, which is toxic to cells [15,16,17,18,19,20]. The detail information regarding the metabolism of SRB, their role in natural processes, or their industrial impacts can be found in the works of R. Rabus, T.A Hansen and F. Widdel [21], J. M. Odom [22] or F. Widdel and F. Bak [23]. Considering the wide range of habitats in which SRB can be found, including their possible impact on human health and economics, it is crucial to study these microorganisms further. The main points that should be investigated are the habitats of this group of organisms, their ecological interactions and finally, the development of a broad range of efficient methods to reliably detect and identify them in the context of both research and clinical usage. The main points that will be a part of this review are shown in Figure 2.

This review aims to describe the ecology and habitats of SRB, their morphology, which can be used in detection via microscopy, selected molecular methods used to identify this group, and finally, microscopic methods for classifying and identifying this very heterogeneous group of microorganisms. These points are intended to provide a complex overview of the techniques which can be used for detecting SRB with an emphasis on their possible impact on health and the economy.

## 2. Ecological Characteristics of Sulfate-Reducing Bacteria

SRB are essential in many ecosystems in terms of number and activity. They occur in fresh and salt waters, soils, mud and sediments, as well as in human diseases or oil fields [24,25,26,27]. They are also present in polluted environments, such as anaerobic parts of wastewater treatment plants, or hot environments like deep-sea hydrothermal springs. By reducing metals and producing hydrogen sulfide, they contribute to the overall biogeochemistry of these extreme environments. At the same time, they are essential elements in the sulfur cycle in nature [2].

### 2.1. Sulfate-Reducing Bacteria and the Sulfur Cycle

Microorganisms have a significant impact on creating and preserving Earth’s environment by participating in biogeochemical cycles, such as the sulfur cycle. Although many transformations of this cycle can occur in a purely chemical reaction, microorganisms significantly accelerate these processes. Sulfur can occur in the environment in all oxidation states, but most of it is found as elemental (S^0^), negatively divalent (H_2_S, R-SH, R1-S-S-R2) and positively hexavalent (sulfate). Each of these forms can only be used by specific organisms, which transform it into a form that other organisms can use [11,28].

The sources of sulfur are abundant on Earth. These are mainly minerals based on sulfate (especially calcium sulfate), sulfide (e.g., ferrous disulfide—pyrite) and elemental sulfur. One can find it in large amounts in the oceans. Sulfur is vital to all organisms; it is necessary for the process of proteosynthesis. While plants and many microorganisms use elemental sulfur and its oxidized states (sulfate, sulfites, thiosulfates), reduce them, and synthesize organic sulfur compounds, animals–including humans–require sulfur in a reduced state. They are, therefore, dependent on plants and bacteria [11]. 

Plants (and some microorganisms) synthesize organic compounds from the sulfate. Those are then digested by animals and broken down by microbes as a substrate. During this decomposition, hydrogen sulfide is formed and released into the environment. In the air, it is oxidized to elemental sulfur, or it is aerobically converted to elemental sulfur or sulfate by colorless sulfur bacteria (thiobacilli and fibrous sulfur bacteria from the genera *Thiothrix*, *Beggiatoa*, etc.). Anaerobically, hydrogen sulfide is oxidized by phototrophic sulfur bacteria, resulting in elemental sulfur (Figure 3).

The opposite process, much rarer in nature, is performed by SRB, such as the genus *Desulfovibrio*, *Desulfotomaculum*, and others. As already mentioned, these microorganisms reduce sulfate under anaerobic conditions to hydrogen sulfide by dissimilatory sulfate reduction [14,20]. Some SRB also use H_2_ in the process of reduction, for which they compete with methanogenic archaea (methanogens) [24,25]. 

Dissimilatory sulfate reduction has several environmental effects [30,31]. For example, sulfide production inhibits the growth of aerobic microorganisms. The formation of sulfides also leads to the binding of free ions of heavy metals (very often, ferrous sulfide is formed, as soluble iron is commonly found in waters and soils). Furthermore, sulfate and H_2_ are utilized (if used by SRB), contributing to anaerobic corrosion of iron, which will be discussed later. In the intestines of animals, dissimilatory sulfate reduction causes the microbiome’s alteration, similarly to waters [11].

### 2.2. Habitats of Sulfate-Reducing Bacteria

It has been shown that SRB encompass terrestrial and aquatic species, freshwater and marine, halophilic, thermophilic and psychrophilic, non-sporulating and sporulating. At the same time, there is great adaptability to different temperatures in these organisms [11]. Their possible microaerophilic nature or at least tolerance to molecular oxygen (O_2_) is also discussed [32]. Habitat (also biotope) is both the biotic and abiotic environment in which a given organism’s species occurs. Interestingly, in addition to the aquatic and soil environment, SRB have also been detected in spoiled food, stomach of ruminants, or intestines of termites [10]. This work will further discuss the distribution of SRB in soils, waters, hot springs and geothermal environments, oil fields, and animals’ large intestines. The following tables (Table 1 and Table 2) show characteristic habitats of selected members of the genus *Desulfovibrio* and other mesophilic SRB.

#### 2.2.1. Water and Water Sediment

Freshwater. In unpolluted freshwater environments, the sulfate concentrations are usually low, around 0.01–0.2 µmol L^−1^; therefore, methanogens predominate over SRB. These conditions are more advantageous for methanogens and allow them to outcompete SRB for a common substrate—acetate and H_2_ [10]. Out of the Gram-negative mesophilic SRB, the genera *Desulfovibrio*, *Desulfobulbus* and also *Desulfoarculus*, *Desulfobotulus*, *Desulfomicrobium* and *Desulfomonile* are mainly represented therein [23]. The sporulating Gram-positive genus *Desulfotomaculum* is often present, occurring primarily in environments with lower salt concentrations [10]. In freshwater and low sulfate environments, SRB can use fermentation and anaerobically oxidize organic matter. For example, the genera *Desulfovibrio* and *Desulfomicrobium* can grow by fermentation of pyruvate to acetate, CO_2_ and H_2_ [1].

Wastewater. Conventional domestic wastewater has sulfate concentrations around 100–1000 µmol L^−1^ with a relatively low proportion of O_2_ due to its lower solubility and rapid consumption by the biological activity of other microorganisms. For this reason, sulfate reduction can be significantly represented here and is responsible for up to 50% of the mineralization of organic matter [33]. Besides, some SRB may survive under microaerophilic conditions and be O_2_ tolerant. Biofilms can be formed, which are very heterogeneous, although only a few millimetres thin. An internal sulfur cycle is formed within the biofilm [34]. 16S rRNA analysis revealed the presence of SRB in wastewater, containing at least six genera of Deltaproteobacteria: *Desulfomicrobium*, *Desulfovibrio*, *Desulfonema*, *Desulforegula*, *Desulfobacterium* and *Desulfobulbus*. Using the Nomar differential interference contrast (DIC) and fluorescence in situ hybridization (FISH) technique, the spatial distribution of SRB was determined. The genus *Desulfobulbus* was found at all biofilm depths, while the genus *Desulfonema* was found mainly on the surface [33].

In general, the interface between aerobic and anoxic environments (up to the biofilm surface) often appears optimal for SRB in wastewater, as more metabolically active microorganisms require more substrate [35]. Quantification (determined with 4′,6-diamidino-2-phenylindole = DAPI) has shown that only 4.8% of the total number of 6.7 × 10^10^ cells cm^−3^ of biofilm were SRB. The genera *Desulfobulbus* (23%) and *Desulfovibrio* (9.4%) predominated. The genus *Desulfobulbus* also degrades propionate to acetate, and *Desulfovibrio* is a significant representative of H_2_-using SRB [33,36]. 

The negative effect of sulfate reduction in wastewaters is toxic hydrogen sulfide production, which is also a source of odor [33,37]. Another disadvantage of SRB′s activity in sewerage networks and water treatment plants is that they cause corrosion of metals, which is a widespread problem in the oil and other industries and will be mentioned later [38]. 

Saltwater. In marine waters, sulfate concentration averages 28,000 µmol·L^−1^ and sulfate reduction, therefore, outweigh methanogens’ activity. The presence of SRB with high metabolic activity is easily recognizable here due to the blackening of the water and sediment, caused by the precipitation of ferrous sulfide, accompanied by the smell of hydrogen sulfide [10]. Sediments of the sea, estuaries and salt marshes, and saline and hypersaline lakes are the most common and the most important SRB habitats in nature due to their high sulfate content [10,11]. 

SRB were also isolated from hypersaline habitats, for example, from the Great Salt Lake (Utah, USA; salinity 24%), the Dead Sea or salt ponds. However, most SRB are adapted to marine salt concentrations or slightly halophilic (with salinity optimum of 1–4% NaCl). Those are representatives of gram-negative mesophilic genera *Desulfovibrio*, *Desulfobacterium*, *Desulfobacter*, *Desulfonema*, *Desulfosarcina* or archaeal *Archaeoglobus*. *Desulfovibrio halophilus* (grows at 3–18% NaCl) and *Desulfohalobium retbaense* (grows up to 24% NaCl), isolated from a hypersaline lake in Senegal, are moderately halophilic SRB [10,36]. In anaerobic marine sediments, sulfate reduction causes up to 50% of organic matter’s total mineralisation in coastal and shelf ecosystems. Sulfate diffuses here up to a depth of several meters. SRB and methanogens do not compete for the substrate but rather complement each other in the degradation of organic matter, forming syntrophic communities. This interaction means that one organism uses the metabolic products of the other. If sulfate resources become limited, SRB may continue to be active due to this association with methanogens [1]. In that case, SRB can fermentatively recover H_2_ from organic matter, which syntrophic methanogens can further metabolize, as it has been observed, for example, in co-cultures of *Methanosarcina barkeri* and the genus *Desulfovibrio* [39].

Hot springs and geothermal environment. Some SRM are thermophilic and grow even in a very warm, geothermal environment. For example, SRM representatives from the domain *Archaea*, genus *Archaeoglobus* (e.g., *A. fulgidus*, *A. profundus*), have been found only in anaerobic, submarine hydrothermal areas and are hyperthermophiles. The genus *Thermodesulfobacterium* (e.g., *T. commune*) has been observed in hot volcanic vents. The optimal growth temperature of these thermophiles corresponds to the environment–for example, the optimum of the genus *Desulfotomaculum* and *Thermodesulfobacterium* is in the range of 54–70 °C, while the growth maximum is between 56–85 °C. The temperature optimum of the genus *Archaeoglobus* is 83 °C and the growth maximum 92 °C [10]. 

#### 2.2.2. Soil and Mud

Soil and mud. Mesophilic SRB also occur in soil. The flooded soil of rice fields, the mud of the riverbanks or seashores all offer favorable conditions for their growth. Although some SRB can survive in the presence of O_2_, long-term aerobic conditions are unsuitable for them. Therefore, in soils, one may find mainly sporulating SRB, for example, the genus *Desulfotomaculum*, which can also fix molecular nitrogen–in other words, it shows diazotrophic growth [11,21]. Examples of diazotrophic SRB are *Dtm. nigrificans* or *Dtm. orientis*. Furthermore, there have also been found non-sporulating SRB species like *Desulfovibrio desulfuricans*, *D. sulfodismutans*, *Desulfobulbus propionicus* and *Desulfobotulus sapovorans* in freshwater mud and *Desulfobulbus marinus* in sea mud. Genera *Desulfobacter* and *Desulfobacterium* have also been detected in the mud [10].

Rice fields. SRB, especially the genus *Desulfovibrio*, specifically *D. vulgaris* and *D. desulfuricans*, have also been detected to a greater extent in regularly flooded rice field soils. Therefore, they are considered important there and can also function as nitrate reducers in this environment [6]. Non-sporulating SRB using lactate, pyruvate or H_2_ as electron donors predominate in rice fields. They occur in clayey clusters of soil deprived of O_2_, in organic residues and mostly in the rhizosphere (near the roots) and spermosphere (on the surface of germinating seeds) of plants. These SRB can even damage rice plants by producing hydrogen sulfide. For example, in rice fields in Senegal, there was a case of severe damage to plants due to the effect of sulfide, while the SRB density of 10^7^ to 10^9^ bacteria per g of dry soil was detected [40].

#### 2.2.3. Oil Fields and Industrial Environment

SRB were also detected in oil fields and reservoirs of crude oil. There, one can find the genera *Desulfovibrio* (*Desulfovibrio longus*) or *Desulfomicrobium* (*Desulfomicrobium apsheronum*) [10]. Thermophilic archaeal SRM (genus *Archaeoglobus*) have been found in deep, warm oil deposits [21]. SRB are often considered undesirable in these environments since their hydrogen sulfide production causes microbial corrosion of tanks or pipes and oil acidification [38].

Microbially influenced corrosion (MIC, also biocorrosion) is a form of interaction of microorganisms with the material’s surface, causing its damage. Microorganisms do not directly cause corrosion but support and accelerate it, leading to significant economic issues. This problem is related not only to the oil industry but also to the gas and water industries [38]. Microbial biofilms may form on the surface of various materials, including metals and their alloys, but even concrete, stone or plastic [2]. As a part of the biofilm, in its deeper layers, anaerobic SRB can grow even in the originally aerobic environment and use the metabolites of the associated microbiota as nutrients while reducing the redox potential of the environment [41]. It is the SRB group, led by the genera *Desulfovibrio* and *Desulfotomaculum*, that is responsible for the most serious forms of microbial corrosion [42].

The occurrence of SRB in oil tanks may be caused by a common operating procedure in the oil industry, used to facilitate the pumping of residual oil from an almost depleted oil field. To do so, water is being injected under high pressure into the tank. If seawater is used, the tank thus obtains approximately 300 mmol·L^−1^ sulfate. Different barotolerant SRB originating from the seawater then oxidises various crude oil compounds while producing hydrogen sulfide [5,42]. Therefore it is known that SRB can anaerobically degrade petroleum hydrocarbons (n-alkanes, xylenes, toluene or ethyl toluene) in the presence of sulfate [43]. Although it is a slow process requiring a chain of unusual chemical reactions, oil is sufficient for them as the only organic matter source. This creates so-called acidic crude oil, which then needs to be further modified to be used [21].

#### 2.2.4. Large Intestine of Humans and Animals

SRB in the large intestine. The presence of the genus *Desulfovibrio* has long been reported in the rumen of ruminants, in the intestines of termites and the intestines of animals and humans [5,44,45,46,47]. SRB are commonly non-pathogenic and can be successfully isolated from fresh human feces [11], but have also been associated with inflammatory bowel diseases, and their significance has been investigated ever since [48]. 

A study by G. R. Gibson [5] found that more than 40% of individuals in the two tested populations (from the UK and South Africa) were colonized by intestinal SRB. The predominant SRB present were classified into the genera *Desulfovibrio*, *Desulfobacter*, *Desulfotomaculum* and *Desulfobulbus*. *Desulfovibrio* was the predominant genus in feces, a genus using mainly lactate and pyruvate. The genus *Desulfobacter* was isolated on a medium with added acetate, which serves as a carbon source. These bacteria oxidizing acetate were more common in the population of South Africa. The same was true for the genus *Desulfotomaculum*, which can use butyrate or valerate and form endospores above that. The use of propionate as a major carbon source has been found in the genus *Desulfobulbus* [49]. Although the occurrence of intestinal SRB was confirmed in both populations, the percentage of methanogens differed in them. Methanogenesis is detectable, among other methods, by the release of methane. In the intestines, fermentation produces H_2_, which can be used by SRB or methanogens [50] for methane production [51].

Gibson’s research mentioned above indicated that the ratio of SRB and methanogens depends on the population’s geographical affiliation and dietary habits. While in 70% of British samples, SRB were detected, in African samples, it was only in 15% [49]. Most of the tested individuals from the United Kingdom did not show a measurable methane content in their breath, so methanogenesis was minimal, and sulfate reduction occurred in their intestines. In contrast, in most South African representatives, methane was measured in the breath, indicating the presence of methanogens. At the same time, minimal sulfate reduction and SRB content was found.

Similar to other environments, there is a competition between SRB and methanogens for a common substrate in the gut. In the large intestine, this shared substrate is H_2_ because methanogens do not use acetate. Due to higher sulfate affinity by SRB, these organisms out-compete methanogens at higher sulfate concentrations [49].

The role of sulfate in the composition of the large intestine microbiome. Sulfate reduction and methanogenesis in the human colon are clearly related to the availability of sulfate. For the most part, their sources are sulfated glycoproteins such as mucin, commonly present in the gut, produced by the intestinal epithelium [5]. However, sulfate compounds can be ingested in varying amounts in the diet, which can affect the composition of the large intestine’s microbiome and, thus, the internal processes–including sulfate reduction and, therefore, the production of hydrogen sulfide, potentially toxic to the intestinal mucosa. Their concentrations were measured by Florin et al. [52] in more than 200 foods (Table 3). A wide range of values was measured; foods most rich in sulfate (>10 µmol g^−1^) include some bread, soy flour, dried fruit, *Brassicaceae* plants or many sausages. Beverages with a high sulfate content (>25 mmol·L^−1^) are beers, ciders and wines. In some foods, sulfate occurs naturally; in others, it is added to the process of sulfurization of foods, carried out to prolong shelf life [52]. 

Christl et al. [51] conducted research on volunteers with methane in their breath, whose diet was changed to have increased sulfate levels for three weeks. Methanogenesis was found to be inhibited in 50% of subjects, and no difference was seen in 50%. The study, therefore, concluded that there are two different types of the microbiome in the large intestines of the volunteers–one group having inactive SRB (in terms of H_2_ utilization), which are activated by a sulfate source and then inhibit methanogens by their metabolism; the other group not having SRB present and therefore uninfluenced by sulfate addition. Another important factor is the presence of other intestinal bacteria, which by their metabolism, facilitate the release of sulfate (for example, by hydrolysis of glycoproteins) and form organic substrates used by SRB [49].

The clinical significance of sulfate-reducing bacteria. Although SRB are considered non-pathogenic, recent studies suggest that they may play a role in developing human diseases. They have been associated with cases of chronic periodontitis [16] and have been isolated from liver and brain abscesses [53], blood and urine [54]. The genus *Desulfovibrio* has also been isolated from the vaginal microbiome of women, further suggesting the possible clinical significance of SRB [31,55,56,57]. Crohn’s disease and ulcerative colitis (UC) are inflammatory bowel diseases (IBD) of humans, which are chronic and of non-specific origin. The influence of SRB on the development of these diseases (especially UC) is being considered. These diseases are still of unclear etiologies, probably depending on environmental, genetic and immunological factors. However, in experiments on laboratory animals, it was found that the microbiome of the intestinal surface plays a significant role in the occurrence of the disease [54].

The possible infectious origin of these diseases has been considered; however, this has not been proven, and patients do not respond to antibiotic treatment. Simultaneously, an abnormally strong immune response of the body against a common commensal microbiome was found in patients with these diagnoses. The administration of probiotics in experimental animals with UC has been beneficial; the application in patients still needs to be empirically verified [58]. It is hypothesized that hydrogen sulfide produced by SRB as a cytotoxic agent may damage the intestinal epithelium, leading to cell death and chronic inflammation [54]. 

Many authors have already revealed that based on the analysis of 16S rRNA sequences, the genus *Desulfovibrio desulfuricans* predominates over other bacteria in patients with active ulcerative colitis [59]. This is shown, for example, by an experiment in which J. Loubinoux [54] used a selective medium to isolate and subsequently identify SRB from the feces of 41 healthy individuals and 110 patients having intestinal inflammation. SRB were detected in 68% of patients with IBD, 37% of patients with other symptoms and only in 24% of healthy people. Using the PCR method, the genus *Desulfovibrio* (*D. desulfuricans*, *D. piger*, *D. fairfieldensis*) was identified in the samples [54]. However, it is not yet clear whether SRB really directly influence the development of idiopathic intestinal inflammations or they just accompany them. For this reason, it is necessary to further characterize the diversity in the morphology of important genera of intestinal SRB.

## 3. Morphological and Biochemical Characteristics of Selected SRB

The group SRM nowadays encompasses about 40 bacterial genera, mainly from the δ subclass of the phylum *Proteobacteria* and three sporulating Gram-positives from the *Bacillus-Clostridium* group of phylum *Firmicutes* and several Gram-negative thermophilic bacterial genera (*Thermodesulfobacterium*, *Thermodesulfovibrio*, *Thermodesulfobium*). To this date, also two hyperthermophilic archaeal species were described, *Archaeoglobus* and *Caldivirga* [60]. More information about the diverse SRM genera can be found in the Bergey’s Manual of Systematics of *Archaea* and *Bacteria*, where all validated genera are covered [61,62]. Due to the above illustrated large number and diversity of genera belonging to sulfate-reducing microorganisms, only bacterial representatives, significantly represented in the intestines of animals, humans and clinical material (including the patients with inflammatory bowel diseases) will be characterized here. The two longest known SRB genera are *Desulfovibrio* and *Desulfotomaculum*. Other SRB often isolated from intestines are of the genera *Desulfobacter*, *Desulfobulbus* or *Desulfomicrobium* [49].

*Desulfovibrio* is the most studied SRB genus and the most common SRB in the intestine [63]. It does not sporulate, is mesophilic (optimal growth temperature 25–37 °C) and may be halophilic [11]. It belongs to the phylum *Proteobacteria*, class *Deltaproteobacteria*, order *Desulfovibrionales*, family *Desulfovibrionaceae* [64]. *Desulfovibrio* is gram-negative, characterized by distinctive mobility through one or more polar flagella. The fresh culture consists mainly of curved rods (also called “comma-shaped”) of 0.5–1.5 × 2.5–10.0 µm. They can also be sigmoid (*D. africanus*) or spiral, rarely also straight–the shape may be influenced by age and environment, thus the medium composition [64]. All members of the genus *Desulfovibrio* contain desulfoviridine, a sulfite reductase, which presence is checked by the desulfoviridine assay–in a positive assay, we see a characteristic red fluorescence after the cell suspension is exposed to light at wavelength 365 nm after adding a few drops of 2 mol L^−1^ NaOH. This test and its morphology distinguish the genus from other SRB. However, a comparative 16S rDNA analysis is needed to distinguish the genera [64]. The type strain is *Desulfovibrio desulfuricans*. It has a curved rod’s shape, or it is sigmoid, with dimensions of 0.5–0.8 × 1.5–4.0 µm and is mobile. *Desulfovibrio piger*, formerly *Desulfomonas pigra* [65], is another species also occurring in animals’ intestines. It has the shape of a rod, is approximately 0.8–1.3 × 1.2–5 µm in size, and is one of the few immobile *Desulfovibrio* species. *Desulfovibrio intestinalis* is a species isolated from termite intestines, has a curved shape and size of 0.4–0.5 × 1–1.4 µm and is motile [64].

*Desulfobacter* is the second most isolated genus from the intestines, also occurring in waters and mud. It belongs to the phylum *Proteobacteria*, class *Deltaproteobacteria*, order *Desulfobacterales*, family *Desulfobacteraceae*. It is a mesophile (optimal growth temperature 28–32 °C), and it does not sporulate [66]. *Desulfobacter* is a gram-negative bacterium, oval to rod-shaped with rounded ends, measuring 0.5–2.5 × 1.5–8 µm. Its shape can also be slightly curved, and some species are pleomorphic. Cells occur individually or in pairs; marine strains tend to form clusters [66]. Motility is provided by one polar flagellum, but some are immobile because the motility may be lost during cultivation. *Desulfobacter* representatives do not contain desulfoviridine as a sulfite reductase, but desulforubidine may be present. The type strain is *Desulfobacter postgatei* with a typical oval shape. Another species, *Dba. curvatus* has vibrioid cells, *Dba. latus* has the shape of large, elongated rods [66]. 

*Desulfobulbus* can be found in aquatic ecosystems, as well as in the stomach of ruminants or manure. It belongs to the phylum *Proteobacteria*, class *Deltaproteobacteria*, order *Desulfobacterales*, family *Desulfobulbaceae*. The name is derived from the word “bulbus”, which means onions, whose shape the cells of *Desulfobulbus* resemble. It is a mesophilic genus with an optimal growth temperature of 25–40 °C. It does not form spores [67]. The genus *Desulfobulbus* consists of gram-negative ovoid rods, sometimes also lemon-shaped with pointed ends, that are 0.6–1.3 × 1.5–3.5 μm in size. Cells can occur individually, in pairs or chains. They are motile by one polar flagellum or immobile. All described species of the genus contain desulforubidine [67]. The type strain is *Desulfobulbus propionicus* with a typical elliptical shape with pointed ends. The more elongated shape can be seen in the species *Dbu. elongatus*, cylindrical shape in *Dbu. marinus* [67].

*Desulfotomaculum* is an SRB genus occurring in the sediments of waters and the intestines, especially of animals. It is one of only three known spore-forming genera of SRB. This makes it the dominant SRB genus in habitats with variable redox conditions (e.g., soil, rice fields). It can be mesophilic (30–37 °C) or thermophilic (50–65 °C). It belongs to the phylum *Firmicutes*, class “*Clostridia*”, order *Clostridiales*, family *Peptococcaceae* [68].

*Desulfotomaculum* is gram-positive, motile utilizing either one polar flagellum or more flagella projecting in all directions (peritrichous). During cultivation, mobility may be lost. They form straight or curved rods 0.3–2.5 × 2.5–15 μm in size, the ends of which may be rounded or pointed. Cells occur individually or in pairs. Endospores are oval to round, with a terminal or central location and cause the cell to arch [68]. The morphology of this genus and especially the formation of endospores makes it easier to distinguish the genus from other SRB. *Desulfotomaculum* also does not contain desulfoviridine or desulforubidine. The type strain is *Desulfotomaculum nigrificans*, which be isolated from water. Other representatives are, for example, *Dtm. alkaliphilum* and *Dtm. acetoxidans*, which can be found mainly in pig manure, or *Dtm. rumini*, which can be isolated from the stomach of ruminants [68].

The genus *Desulfomicrobium* belongs to the phylum *Proteobacteria*, class *Deltaproteobacteria*, order *Desulfovibrionales*, family *Desulfomicrobiaceae* [69]. This bacterial genus is non-sporulating. It is mostly mesophilic (optimum 25–30 °C), but a thermophilic species *Desulfomicrobium thermophilum* has also been discovered in a hot spring in Colombia [70]. *Desulfomicrobium* sp. occurs mainly in sediments and mud, but furthermore, the species *Desulfomicrobium orale* has also been isolated from periodontal vesicles in patients with periodontitis [71]. These bacteria are gram-negative, short and straight or ellipsoidal rods with rounded ends of 0.5–0.9 × 1.3–2.9 μm in size. The cells occur individually or in pairs and are motile by one polar flagellum. Physiologically, the genus is similar to the genus *Desulfovibrio*, but its cells do not have the curved shape of *Desulfovibrio,* and they also lack desulfoviridine. Therefore they can be easily distinguished by the desulfoviridine test [72]. The type strain is *Desulfomicrobium baculatum*; other examples are *Desulfomicrobium apsheronum* or *Dsm. norvegicum* and the already mentioned *Dsm. thermophilum* and *Dsm. orale* [72].

In summary, SRB are a diverse group of microorganisms that share the ability to reduce sulfate. From the morphological point of view, they occur as curved rods, straight rods and spirals of various sizes, the shape often affected by age and the environment. The genera found in the intestines tend to be mesophilic and usually well motile. Some SRB (genus *Desulfotomaculum*) form spores and are therefore very resistant to adverse conditions. Although today’s microbiology often uses molecular methods based on nucleic acids, microscopic observations are still very useful, and so is the knowledge of the morphology of the individual genera. By combining both approaches, we achieve the best results and comprehensive knowledge of the investigated microorganism.

## 4. Molecular Methods for Detection of Sulfate-Reducing Bacteria

Molecular methods analyze DNA, RNA, or proteins found in the microorganisms and are used in microbial ecology to study the diversity and abundance of microbes in detecting and identifying specific strains. The entire phylogenetic system of microorganisms is now based on comparing the information content of these molecules [73].

### 4.1. Molecular Methods for Classification

Taxonomy classification methods are used to group organisms so that the new isolate can later be more easily characterized by comparison with known microorganisms. 

For a long time, the “gold standard” of prokaryotic taxonomy has been DNA-DNA hybridization (DDH), reflecting the relatedness between two genomes. According to DDH, two microorganisms belong to the same species if the DNA-DNA similarity is higher than 70% or if the hybrid DNA duplex differs by melting point (ΔT_m_) by 5 °C or less. The result of DNA-DNA hybridization is usually in accordance with phenotypic, chemotaxonomic and other data [74]. Advances in technology nowadays allow us to perform DDH in silico [75], too. The consensus among taxonomists is that all relevant taxonomic information about a microorganism is incorporated into its genome sequence [76]. Genome sequence-derived parameters, for example, the average nucleotide identity (ANI) of common genes, are a robust measure of the genetic and evolutionary distance between species [77]. Therefore, alternatives based on the genomic sequence similarity are being introduced, and a whole-genome sequence is now required to describe a species officially.

Phylogenetic analysis of 16S rRNA genes is used more often than DDH, as it is less demanding in terms of implementation and finances. Two microorganisms are considered different species when their sequence identity of the 16S rRNA gene is lower than 97%. When the percentual identity is higher, they can be considered related, and DNA-DNA hybridization should be performed to prove their belonging to the same species [78]. 16S rRNA genes are highly conserved in prokaryotes and may also contain variable regions [79]. 

The small subunit of the ribosome in prokaryotes is considered a “molecular clock”, based on the assumption that the rate of mutations in molecules is constant over time. Thus it is possible to determine the approximate evolutionary distance between two species. However, different genes have different rates of mutation, so this concept can be considered rather simplified. For some cases, genes encoding various specific proteins–functional genes are also used [79].Similarly, another specific nucleic acids (NA) regions may be used, such as the 16S-23S rDNA internal transcribed spacer (ITS). This DNA region is used to differentiate closely related species, as 16S rDNA changes so slowly during evolution that it is almost impossible to differentiate some microorganisms by their sequence [73]. 

The PCR (polymerase chain reaction) method is used to analyze these genes and NA regions. With the development of the rRNA genes sequence analysis, significant advances have been made in the taxonomy and phylogeny of the very diverse SRB groups [80]. Complete sequences of genomes of many members of the *Bacteria* are known, as well as several *Archaea*, whose genome is often smaller. Based on the rRNA sequences, four SRB groups were created: gram-negative mesophilic, gram-positive sporulating, thermophilic bacterial and thermophilic archaeal [81]. Six groups were further identified within the domain *Bacteria*, according to the rDNA analysis (Table 4) [82].

New species of SRB were also discovered by 16S rDNA analysis. For example, in 2001, a new species *Desulfomicrobium orale* was found in the periodontal pockets of the patients suffering from periodontitis. Thus, this species could be distinguished from other members of the genus *Desulfomicrobium* using a fatty acid profile [71].

In contrast, *Desulfovibrio piger* was formerly classified in the genus “*Desulfomonas*” as its only species *“Desufomonas pigra”.* It was reclassified into the genus *Desulfovibrio* based on its growth characteristics, percentage of G+C bases, 16S rDNA sequence and 16S-23S rDNA ITS sequence [65]. In 2017, this species’ complete genome sequence was obtained, specifically of a strain isolated from the feces of a patient diagnosed with ulcerative colitis, *D. piger* FI11049 [83]. 

### 4.2. Molecular Methods for Detection and Identification

The identification of SRB in clinical laboratories by classical methods is challenging because, for example, the genus *Desulfovibrio* predominant in clinical material [84,85], has slow growth and can be displaced by other microorganisms during its growth [54]. For this reason, special media with sulfate and iron are used for the isolation of SRB, and then faster molecular methods can be used [4,86]. 

Molecular techniques for identifying bacteria are DNA-based or phenotypic. The method based on the similarities of the primary structure of DNA is the DNA-DNA hybridization mentioned earlier; phenotypic methods are often based on PCR reactions, in which random sequences or, vice versa, only specific sequences of the bacterial genome are amplified. Random amplification of polymorphic DNA by PCR (RAPD-PCR) or arbitrarily primed PCR (AP-PCR) are techniques based on the amplification of random sequences. In these methods, one short (10–20 bp) primer is used that randomly links to more specific sites on the DNA. During analysis by gel electrophoresis, specific fingerprints are being formed, which can be used for identification at the species to strain level when comparing with a library [87]. These two methods are relatively sensitive and inexpensive and have been used, for example, to investigate genes responsible for the resistance of some SRB to metals, but have also yielded false-positive results or formed chimeras composed of rRNA and mRNA [88].

Amplified specific sequences are, for example, enterobacterial repetitive intergenic consensus sequences (ERIC-PCR) or repetitive extragenic palindromic sequences amplified by PCR (REP-PCR). The results of different techniques are often combined, and it can be determined whether clusters of similarity form between species [78]. 

Versalovic was the first to investigate the distribution of repetitive DNA sequences in the *Eubacteria* domain. He began using primers complementary to the REP and ERIC DNA sequences for PCR. The result of his work was the detection of species and strain-specific DNA traces (fingerprints) for many gram-negative eubacteria, which can be used for their faster identification [89]. These sequences are commonly found in several copies in the genome of gram-negative bacteria. REP sequences are 35–40 bp long, ERIC sequences measure 124-126 bp, and are located within bacterial genes [90]. 

Based on these findings, SRB bacterial typing, specifically of the species *Desulfovibrio desulfuricans*, was performed. Bacteria from the environment (soil), human feces and appendix biopsy samples were examined. Characteristic genomic profiles for different strains of *Desulfovibrio desulfuricans* were obtained by both REP-PCR and ERIC-PCR. However, some of the strains did not contain these sequences. Analysis of similarity clusters revealed that soil isolates show greater heterogeneity than intestinal isolates, which may be due to the intestines′ very specific environment. This method can be further used for bacterial typing and differentiating strains of this genus [90]. Later, the rrn operon in fragments encoding the 16S and 23S rRNA genes and the ITS spacer were analyzed in *D. desulfuricans* strains. Also, REP-PCR, ERIC-PCR and AP-PCR were performed. These methods are useful for determining similarities between strains as well as for distinguishing them. The REP-PCR method had the highest discrimination power [59]. 

Due to the fast development of molecular biology, many oligonucleotide probes and PCR primers for SRB have been developed in recent years, targeting specific sequences, particularly 16S rDNA. These probes and primers can be used to detect and identify SRB, especially in mixed populations, both in the environment and elsewhere [82,91]. Using multiplex PCR, which utilizes several different primers simultaneously and thus amplifies several different sequences simultaneously, Loubinoux, Bronowicki and their colleagues detected SRB in 151 fecal samples from healthy individuals and patients with intestinal inflammation. The species found were *D. piger,* “*D. fairfaieldensis*” and *D. desulfuricans,* with the incidence of *D. piger* being slightly higher in individuals with ulcerative colitis or Crohn’s disease [54]. Using molecular techniques, a representative of the genus *Desulfovibrio* was also identified, obtained from an isolate from a patient’s blood with peritonitis and appendicitis [92]. Upon determining the microscopic morphology and performing the Gram staining, determining the sensitivity to antibiotics and the culturing and biochemical tests, everything referred to the fact that bacteremia was caused by the genus *Desulfovibrio* [93]. PCR and complete sequencing of 16S rDNA was performed. The sequence was compared with the GenBank bioinformatics database, showing 99.9 % agreement with the sequence of “*Desulfovibrio fairfieldensis*” previously obtained from a liver abscess of an 85-year-old man [92,94].

An important and best-studied functional marker for sulfate reducers is the drs-gene, encoding the dissimilatory sulfite reductase (DSR, EC 1.8.99.1). This enzyme catalyzes the reduction of sulfite to sulfide and is required by all sulfate-reducing microorganisms [95]. It was used as a marker in many studies targeting SRB employing cloning and sequencing of dsr gene libraries or by the terminal restriction fragment length polymorphism (TRFLP) method [96]. A drsB-based denaturing gradient gel electrophoresis (DGGE) was also developed to assess the composition of environmental SRB communities [95].

These PCR-based molecular techniques are often used to classify, detect, and identify various microorganisms, SRB included. On the other hand, methods not requiring the NA amplification are also available, for example, flow cytometry, microarrays, or FISH, which will be described later. These days, also metagenomics studies are rapidly emerging.

### 4.3. Usage of Metagenomics

Since the late 1990s, metagenomics emerged as a tool for studying the sample’s overall microbial genetic material. It made many unculturable microorganisms detectable and allowed for identifying various novel enzyme activities and protein structures. Methods based on generating metagenomics DNA libraries are used the most; however, massive metagenomic sequencing projects are taking place. Finally, preparing metagenomic expression libraries used for finding enzyme activities of interest can be performed [97]. The usage of metagenomics for environmental samples has already revealed novel enzymes from metagenomes from marine environments, soils or extremely acidic or alkaline environments [97]. In several studies, interesting traits of some SRB were revealed–for example, their higher affinity to bioplastic materials in the marine environment compared to other microorganisms, accompanied by the formation of biofilms and presence of enzymes responsible for bioplastic degradation. The responsible genomes were identified as novel species of genus *Desulfovibrio* and *Desulfobacteraceae* or *Desulfobulbaceae* family [98]. 

A metagenomics study combined with other methods has also identified a potentially H_2_-consuming SRB in anoxic coastal sediments, particularly a member of the order *Desulfobacterales*. Molecular hydrogen is the main intermediate of anaerobic carbon mineralization in marine sediments; its scavenging is essential to keep anaerobic degradation energetically favorable. Although this has been long known, the exact SRB consuming H2 in situ was not discovered until this study. These SRB also play a role in suppressing methane formation and its release from sediments [99].

Another work based on metagenomics also identified putative novel *Deltaproteobacteria*-related SRB genera, found in revegetated acidic mine wasteland. This indicates that there is still a need for more studies of the taxonomic diversity of SRB [100]. However, for gut microbiota samples, few metagenomic experiments have dealt with SRB so far. Furthermore, most of the intestinal microbial diversity studies, including SRB, were still performed by PCR-based methods and mainly on animal guts, especially ruminants, for example, [101]. As mentioned earlier, mainly 16S/18S rDNA amplification-sequencing techniques, DGGE, TRFLP or similar methods were used for these studies [102,103,104]. Therefore, little is known about these communities’ functional diversity, which can be changed by using the activity-based metagenomic approach. For example, Ferrer et al. have prepared a metagenome library of bovine rumen microflora, searching for novel hydrolase diversity. Sequence analysis of the retrieved enzymes revealed several new genes coding enzymes with hydrolytic activity, while two of them were assigned by bioinformatic analyses to members of SRB–particularly to *Desulfovibrio vulgaris* subsp. *vulgaris* str. Hildenborough and *Desulfotalea psychrophila* [105]. Thus, we can say these SRB are probably part of a rumen community, assisting the grazing animals with the digestion of plant mass constituents. 

Although studies mentioned above illustrate well the immerse possible impacts of the sulfate-reducing metabolism on the environment, the usage of metagenomics in studying the SRB diversity in animals’ intestines and especially humans can be considered at its beginnings. Hopefully, the metagenomics approach will soon bring interesting new data to science.

## 5. Microscopic Methods for Detection of Sulfate-Reducing Bacteria

The history of microscopy dates to the 13th century, when Robert Bacon tried to observe small objects through drops of water or glass. Later, in the 17th century, the Janssen brothers assembled the first composite microscope and thus laid the foundation for contemporary microscopy. Since then, there has been a striking increase in the possible magnification of microscopes, and the occurrence of image distortion aberrations has been minimized. Due to the complexity of microscopic techniques and current types of microscopes, only some important variants and those that can be useful in SRB research will be described.

### 5.1. Light Microscopy

The design of the light microscope can be either conventional or inverted, where the sample is inserted from above. In both cases, the sample is placed between the light source and the lens. Depending on the illumination mode, which is essential for image contrast, various modified techniques of this microscopy have emerged, including fluorescence microscopy. The basic and oldest microscopic technique is observation in the bright field. The bright field’s disadvantage is low contrast, which must be increased, for example, by staining. It can also be increased using modifications of this method, which will be described below. Contrast can also be increased using this method’s modifications, which will be described below.

### 5.2. Basic Light Microscopy Modifications

For the initial work with bacterial cultures, light microscopy in various modification can be used. For example, for determining the growth and checking the presence of the bacterial cells of interest, for example, the SRB, some authors use phase contrast microscopy [106]. This method is based on visualizing the phase shift of the light, diffracted by the sample by its interference with suitable non-diffracted radiation. This allows us to see even uncolored objects. The possible occurrence of unwanted “halo” phenomenon can be avoided, for example, by using the Differential Interference Contrast (DIC) (also called Nomarski interference contrast—(NIC)) [107]. Also, dark field microscopy, which is the opposite method to the bright field illumination, can be used to visualize any bacterial cells in an uncolored sample. This way, we can see even very small objects, although the contrast of the contours produced by this method is excessive and not very realistic. Suppose the individual parts of the diaphragm are replaced by transparent color filters. In that case, we obtain the Rheinberg illumination, which further increases the observed object’s contrast by being color-selective [108].

### 5.3. Fluorescence Microscopy

Towards the end of the 20th century, with the development of electronics and the invention of CCD cameras (Charged Coupled Device), evaluating microscopy samples changed from ocular observation to image processing via a connected computer, facilitating the analysis of small samples. Likewise, efforts continued to increase the microscope’s resolution by shortening the wavelength of light used. A microscope for UV radiation was constructed, which was used in the subsequent construction of a fluorescence microscope, allowing the visualization of almost arbitrarily small details and a very small number of molecules. Later, a confocal microscope was assembled, significantly improving contrast and resolution when observing thicker biological samples [109]. The fluorescence microscope is arranged similarly to a dark field observation microscope. However, it is adapted to use the fluorescence phenomenon by adding a dichroic mirror and a pair of filters.

The fluorescence microscope design allows us to observe structures that are not visible by other methods already mentioned. Therefore, it finds application mainly in natural sciences and medicine. In microbiology, it is used to identify microorganisms, determine their abundance (e.g., by using DAPI), observe pathogenic microorganisms in a mixture, or detect the presence of a specific NA sequence in cells. 

The method is based on the phenomenon of fluorescence–after irradiation with light of a certain wavelength λ_EX_, some substances absorb the energy of this radiation (excitation) and soon after, they emit light of another wavelength (λ_EM_) again (emission). There are three phases taking place from a physical point of view, described by the Jablonski diagram (Figure 4) [109].

In the 1st phase, the energy of the incident photon from the light is absorbed by the fluorophore, and its electrons are excited, moving onto a higher energy level (excited electron singlet, S_1_’). In the 2nd phase, lasting only 1–10 ns, the fluorophore changes its conformation and energy is dissipated–a part of it is emitted out in the form of heat, another part in the form of a photon. The electrons are in the relaxed excited electron singlet S_1_. In the 3rd phase, a photon is emitted from the fluorophore, and its electrons return to the initial energy level S_0_. Due to the loss of energy during the 2nd phase, the emitted radiation’s energy is lower, and its wavelength is higher than the excitation one. The intensity of radiation is also lower. The difference in energies and wavelengths (λ_EX_–λ_EM_) is called the Stokes shift [110]. The sample can either show autofluorescence (e.g., chlorophyll, keratin, some cofactors of enzymes, some amino acids, vitamin A, have a natural fluorescence ability), or we can bind it to a fluorescent dye, a fluorophore (also fluorochrome), making the sample visible in a fluorescence microscope.

Widely used fluorescent labels are, for example, fluorescein isothiocyanate (FITC) or tetramethylrhodamine-isothiocyanate (TRITC), particularly for fluorescent labeling of proteins. Fluorescent probes used to label nucleic acids include, for example, DAPI, Hoechst 33342 and 33258, propidium iodide, SYTO and others. Cyanine dyes Cy3 or Cy5, or many Alexa Fluor variants, are also popular as fluorophores for their higher fluorescence intensity. Different fluorophores have different specific fluorescence spectra–comprising the wavelengths at which they can be excited and in which the radiation is emitted (Table 5).


One of the filters (excitation filter) of the epifluorescence microscope defines the wavelength of the required incoming excitation radiation; the other (emission, barrier filter) is placed behind the lens and transmits only the emitted radiation. The semipermeable dichroic mirror reflects excitation radiation through the sample lens to the sample. It then transmits to the ocular only the emitted radiation of longer wavelength while removing high-intensity radiation harmful to the eye. The dichroic mirror and filters are inserted into a cube-shaped body in the fluorescence microscope, forming a so-called fluorescent filter block. Filters are assigned to the respective fluorophores based on the wavelength of the radiation they transmit and vice versa [109]. 

Fluorescence is also used in other techniques than microscopy. With the help of specialized devices, cell counters, it is possible to count easily and quickly all the fluorescently labeled microorganisms present and determine their viability. Analysis of various cell experiments can also be performed using a fluorimeter. With fluorimeter, we can determine the viability of cells but also the activity of enzymes and examine metabolites and proteins of organisms. Fluorescence is also used in flow cytometry, used for specific cell separation and characterization [110]. Fluorescence microscopy can be used for fixed slides to examine specific cells. Using fluorophores, we can detect the presence of a specific group of microorganisms or a specific nucleic acids sequence in a sample (both from the environment and clinical material) by FISH, a method described below. In clinical laboratories, a fluorescence microscope is used, for example, to detect the infectious agent of tuberculosis or to detect fungal infections, and it allows for reading the results of immunofluorescence assays.

#### 5.3.1. DAPI Fluorescent Dye

DAPI, 4′,6-diamidino-2-phenylindole, is a widely used fluorescent probe. It binds highly specifically to double-stranded DNA regions rich in AT pairs. After binding to DNA, its fluorescence intensity increases up to twenty times because the fluorescence ceases to be quenched by water molecules. The maximum excitation wavelength of DAPI in complex with DNA is 358 nm (UV radiation), the maximum emission wavelength is 461 nm (visible blue light). DAPI also binds to RNA to some extent, but by a different mechanism, with an emission maximum of about 500 nm. This fluorophore stains all DNA in the sample. At the same time, the observation of inorganic particles is excluded. It penetrates through the intact membrane of living cells, but the staining is more effective in the fixed cells. However, this also means that DAPI needs to be handled carefully. The observable result is blue-stained bacterial cells on a dark background. An example of the DAPI usage can be seen in the Figure 5 below. This was performed experimentally by one of the authors and used here to better illustrate the technique, with no other intentions. It was performed by filtering the diluted sample onto a Millipore membrane filter after separating the cells from the rest of the sample with detergent and subsequent centrifugation. DAPI was applied at a concentration of about 65 µg mL^−1^ to dried filters and incubated for 10–15 min in a refrigerator. Afterwards, the samples on the filters were washed in Milli-Qwater (MQ), in ethanol and finally again in MQ water. The filters dried on air were then placed on a glass in a drop of immersion oil, covered with another drop of oil and covered with a coverslip [111]. The evaluation was performed with a fluorescence microscope using an immersion objective.

DAPI is used mainly in microbial ecology to determine the total abundance of microorganisms and their density. Similarly, we can use propidium iodide, SITO, and so forth. DAPI staining can be combined with phylogenetic staining (FISH methods), which allows the detected organisms not only to be quantified but also to be assigned to a specific species or phylogenetic group. We can also measure cell size, calculate biovolume (cell volume) and total biomass, which is crucial in ecological studies dealing with carbon fluxes in food chains or biogeochemical processes [112].

The analysis of biofilms from SRB-containing environments usually begins by determining the total abundance of microorganisms using DAPI fluorescence staining. Especially in the water, oil and gas industries, it is necessary to consider the great microbial diversity in the given environment, which cultivation techniques usually significantly skew. To prevent economic losses by microbial corrosion or acidification of oil, it is necessary to monitor the state and presence of different groups of microorganisms in, for example, tanks and pipes [113]. The total number of microorganisms per unit of volume or area can be determined by the DAPI method. Then fluorescence labeling with an oligonucleotide probe for SRB (see FISH method below) can be performed. From these numbers, the percentage of SRB or even directly targeted species can be calculated [114,115]. Using electron scanning microscopy, the present microbial population’s appearance and distribution can be further examined [115]. Molecular methods using PCR can then determine the exact species or their corrosive activity in the sample [113].

#### 5.3.2. Fluorescence In Situ Hybridization

Fluorescence in situ hybridization (FISH) is an important method used in biology, microbiology, and microbial ecology. It combines a molecular approach and a fluorescence microscope; it can also be used in flow cytometry. Unlike most molecular techniques, it does not require nucleic acid amplification. It was the shortcomings and errors occurring in PCR that gave rise to the development of PCR-independent methods. FISH is a fast, sensitive and specific method that does not require cultivation. 

FISH enables visualization of the targeted nucleic acid sections directly in the cell, in situ, using a fluorescently labeled DNA or RNA probe. It can be used to detect, identify, and characterize microorganisms in a mixed sample and study the distribution of microorganisms in biofilms. In clinical laboratories, FISH is used to detect species with little activity or yet un-isolated, which is responsible for the disease. The principle of FISH is based on the ability of single-stranded NA molecules to form hybrid molecules with each other based on complementary sequences. The NA probe is thus able to bind to the targeted nucleic acids of the sample (Figure 6). 

The probe is an artificially designed, fluorescently labeled oligonucleotide, an NA region of 20–30 nucleotides in length, complementary to the sequence of a conserved region of nucleic acids of the target microorganism. Often, probes target highly conserved rRNA of the small subunit of the ribosome (16S), or even its large subunit (5S and 23S). In addition, ribosomes provide the advantage of signal amplification because there are relatively large amounts in the cell. The probes can be both species-specific and functionally specific, detecting specific metabolic processes. Before hybridization, they are labeled by nick translation, random primed labelling or PCR. Probes can be labelled with fluorescent dyes (directly) or modified nucleotides containing a hapten (indirectly). If indirect labelling is used, an additional step is added–visualizing the hapten by enzymatic or immunological detection [116]. NA hybridization will only take place if the NA is single-stranded. Double-stranded nucleic acids must first be denatured using heat and, for example, formamide. Optimal temperature, ion concentration, and hybridization duration are required to create stable hydrogen bonds between the probe and the NA. Lower temperature and ion concentrations facilitate the hybridization; hydrogen bonds formed are then stronger. However, if the temperature or ion concentration was too low, hybridization might occur between molecules that are not exactly complementary. Thus, for reliable hybridization, the conditions must be sufficiently stringent and also the sample is being washed with a buffer after the hybridization to remove poorly bound probe molecules. The FISH method’s procedure is generally following–firstly, modification of the sample takes place–its permeabilization and fixation, followed by filtration of the cells through membrane filters. The NA in the cells is then hybridized to a complementary, fluorescently labeled probe (for several tens of minutes to several hours, incubated in heat). Afterwards, samples are washed with buffer and distilled water. Finally, slides are stained with DAPI, placed in a mounting medium and visualized using a fluorescence microscope [116].

FISH has existed for more than 25 years and has become a standard tool used in microbial ecology, as, unlike the PCR-based methods, it allows to analyze their spatial distribution. The method in its original variant, targeting rRNA of certain groups of microorganisms, is especially suitable for identifying actively growing cells in a nutritionally rich environment [117]. Due to the availability of a large amount of sequence information (mainly 16S rRNA sequences of culturable but also environmental microorganisms) in public bioinformatics databases, many different probes can be prepared, targeting different groups or genera of *Bacteria* or *Archaea*, including various members of SRM [118].

#### 5.3.3. Fluorescence In Situ Hybridization in SRB Studies

As already mentioned, SRB are often found in biofilms. Considering that biofilms can be a problematic target for the treatment in case of an infection, it is of more and more importance to study them. For studying SRB, the FISH method was of considerable importance, for example, in the visualization of a newly discovered microbial consortium catalyzing the anaerobic oxidation of methane to CO_2_ in marine sediment. The data indicated an interaction between *Archaea* and SRB, with a cluster of *Archaea* representatives surrounded by a layer of SRB. Both spatial distribution and identification of the microbial consortium were performed by FISH using a 16S rRNA probe. *Archaea* were primarily targeted by the EelMS932 probe, the SRB by the probe DSS225 and DSS658, specific for the SRB development branch *Desulfosarcina*/*Desulfococcus* [119]. Using the FISH method, it has also been found that the genus *Desulfobulbus*, highly metabolically adaptable and evenly distributed over the entire width of the biofilm, predominates in the wastewater biofilms of the SRB group, as mentioned earlier [34]. Similarly, the combination of FISH and rRNA slot blot hybridization allowed authors of another study to identify SRB members and calculate specific cellular rRNA contents with respect to its localization in the marine Arctic sediment [119]. Last but not least, another interesting study combined FISH with PhyloChip, quantitative PCR targetting drs-genes and other methods, revealing selective enrichment of SRB, present in archaea-dominated subsurface biofilm, forming a consortium in a sulfidic spring [120].

#### 5.3.4. Modifications of the FISH Method

Today, FISH has numerous variants and modifications that seek to increase the sensitivity of this method. Of course, it is necessary to carefully select a suitable probe sequence and test and optimize the method. The problems to be solved can be insufficient cell permeability or a small number of ribosomes (usual targets of the FISH probes) in metabolically inactive cells [121]. To increase the signal intensity, it is possible to use multiple probes (1–5) targeting different sections of the 16S rRNA or auxiliary oligonucleotides to facilitate probe binding. Polynucleotide probes (longer than 50 nucleotides) or peptide-nucleotide probes (PNA) can also increase the assay’s specificity. In environmental samples, the antibiotic chloramphenicol can stop proteosynthesis, rRNA degradation and, above all, arrest cell division, therefore leading to the accumulation of rRNA in the cell and signal amplification [121]. One of the FISH modifications is, for example, the Multiplex FISH, which makes many DNA sequences visible at the same time, or Flow-FISH is used in flow cytometry. A relatively new method is GOLD-FISH, combining FISH with scanning electron microscopy, allowing very high resolution [122]. Some of the interesting FISH modification for usage in SRB studies will be described below.

Microautoradiography in combination with FISH (MAR-FISH)**.** The FISH method modification used in SRB study, especially to study their ecology, is microautoradiography in combination with FISH. It is a polyphasic approach to analysis, where microautoradiography provides information on microorganisms’ metabolic activity, and FISH allows the identification, localization, and quantification of microorganisms, for example, in activated sludge [123]. A radiolabeled substrate is added to the sample, and its uptake is monitored. The most used substrates are glucose, acetate and different amino acids, as they provide a general view of the diversity of metabolisms in an environmental sample. Labeling is performed, for example, with ^3^H, ^14^C, ^35^S, or ^33^P isotopes [124]. The taxonomic identification obtained by FISH gives one an idea of how and which cells of the community use a given substrate; thus, we can obtain information about the existing food chain [121]. The substrate used is made visible using a light-sensitive autoradiographic emulsion containing silver halide crystals. During the exposure (taking 12 h to several days), electrons are released from the radioactive isotopes, which strike the crystals and form a so-called “latent image”. Subsequent development in the developing agent produces a true image, and the emulsion crystals precipitate in the form of black grains inside or near the cell using the substrate. They can be observed with a fluorescence, confocal or transmission electron microscope [125]. 

SRB can thus be detected using labeled sulfate and setting anaerobic conditions. The MAR-FISH method is used particularly for the analysis of SRB present in biofilms in wastewaters [35]. MAR-FISH is also used to study the previously mentioned microbial corrosion. To prevent and solve this problem, it is very important to analyze the biofilm structure (identify the occurring organisms), the spatial distribution of microorganisms in it and the ongoing metabolic processes in situ [33]. The previously known measurement by microelectrodes, which can be used to determine microbial activity in situ based on the changes in the substrate’s chemical profile, did not have sufficient resolution to distinguish information about the uptake of the substrate by individual cells. Analysis by MAR-FISH made this possible [124].

Tyramide signal amplification FISH (TSA-FISH). Another variant of FISH that can be encountered in detecting microorganisms, including SRB, is tyramide signal amplification FISH, also known as catalyzed reporter deposition FISH (CARD-FISH). It can be combined with MAR-FISH, too. This method uses horseradish peroxidase (HRP), an enzyme that catalyzes the fluorescently labelled tyramide’s oxidation in the cell, used to label the probe. Tyramide becomes an active free radical and binds to the surrounding electron-rich proteins. Tyramide-dye complexes on proteins then significantly enhance the fluorescence signal. Signal amplification using this method is relatively fast; it does not lower the resolution and allows for background color reduction and multiple detections [126]. It increases the fluorescence signal up to 12 times compared to conventional probes. The disadvantage is the need to treat the sample with lysozyme or the like to allow a relatively large HRP molecule to penetrate the cell–lysozyme can damage the sample. TSA-FISH can be targeted to both rRNA and mRNA or tmRNA (transfer-mediator RNA), which is present for example in the genus *Desulfotomaculum* [121]. This method is used to analyze environmental samples, such as freshwater and marine waters, sediments and soil [127]. It is also widely used in histology and cytochemistry; therefore, it could be used to detect SRB in clinical material. The scheme of this method is shown in Figure 7.

To summarize, although some light microscopy methods (Gram staining, phase contrast, etc.) are undemanding, they usually require more time and require good preparation of the specimen and the researcher’s experience. Therefore, they are somewhat neglected in the shadow of faster molecular methods. However, they enable the visualization of microorganisms and their communities, morphological characterization, bacterial viability and possibly their more accurate identification. Efforts to constantly increase the resolution of the microscope gave rise to fluorescence, confocal and electron microscopy. Fluorescence microscopy is used to determine the abundance of microorganisms, for example, using the dye DAPI and the FISH method. FISH does not require the cultivation of the sample and is excellent for studying microorganisms from the environment, including those in biofilms, and is thus well suitable for the detection of SRB in various habitats.

## 6. Conclusions

Sulfate-reducing microorganisms are a group of anaerobes with a unique metabolism. They are found in almost all environments, are highly adaptable and play an indispensable role in the sulfur cycle on Earth. They comprise genera from both domains, *Bacteria and Archaea*. They include aquatic, soil and intestinal species. Environmental SRB have been found in rice fields or soil on the banks of rivers and seas. In the freshwater environment, they are significantly represented, especially in wastewater, while saltwater is their most common and most important habitat. A significant disadvantage of sulfate reduction in wastewaters is the increased hydrogen sulfide production, which can promote corrosion of materials. SRB are also the cause of undesirable crude oil acidification. The microbial biofilms forming in pipes and tanks are very resistant and allow SRB to survive even in formerly aerobic conditions. For this reason, it is important to thoroughly study biofilm composition and internal chemical cycles to find appropriate prevention and treatment. SRB have also been isolated from the periodontium or in samples of various human diseases, and they also occur in the large intestine. In the intestines of animals and humans, SRB compete with methanogens for a common substrate, and sulfate reduction (or, conversely, predominant methanogenesis) is related to the availability of sulfate, which enters the intestines with various foods. In addition, the genus *Desulfovibrio* has been shown to predominate over other bacteria in patients with ulcerative colitis. It is, therefore, necessary to further characterize the physiology, genetics and morphology of intestinal SRB. 

Although today’s microbiology makes extensive use of molecular methods, microscopic observations can still be very useful. Molecular methods differ depending on whether we use them to classify microorganisms or for identification in a clinical laboratory, research or in microbial ecology. The methods used in prokaryotic taxonomy are mainly DNA-DNA hybridization and analysis of 16S rRNA genes of microorganisms. Although molecular methods do not require the cultivation of microorganisms, they are often more expensive and can produce errors. In addition, they usually do not provide information on the structure and distribution of the microbial community. Thus, combining these methods with microscopy, such as fluorescence and the FISH method, is eligible. Microscopic methods enable the visualization of microorganisms and their communities, morphological characterization, assessment of bacteria’s viability, or their identification. Fluorescence microscopy offers a higher resolution than conventional light microscopy, and it is used to determine the abundance of microorganisms, for example, using DAPI dye and/or the FISH method. FISH combines a molecular approach with microscopy and does not require the cultivation of microorganisms. Therefore, it is excellent for the study of microorganisms from the environment, including those in biofilms. FISH and its variants, such as MAR-FISH, are also well suited for detecting SRB, especially in biofilms of wastewater and seawater, in reactors, on materials prone to microbial corrosion or in mixed cultures from the intestines of animals.

Both molecular and microscopic methods have their advantages and disadvantages. Their combination is ideal for obtaining comprehensive information and understanding of the diversity of SRB in a variety of environments in nature and the large intestine of animals, including humans.

## Figures and Tables

**Figure 1 ijms-22-04007-f001:**
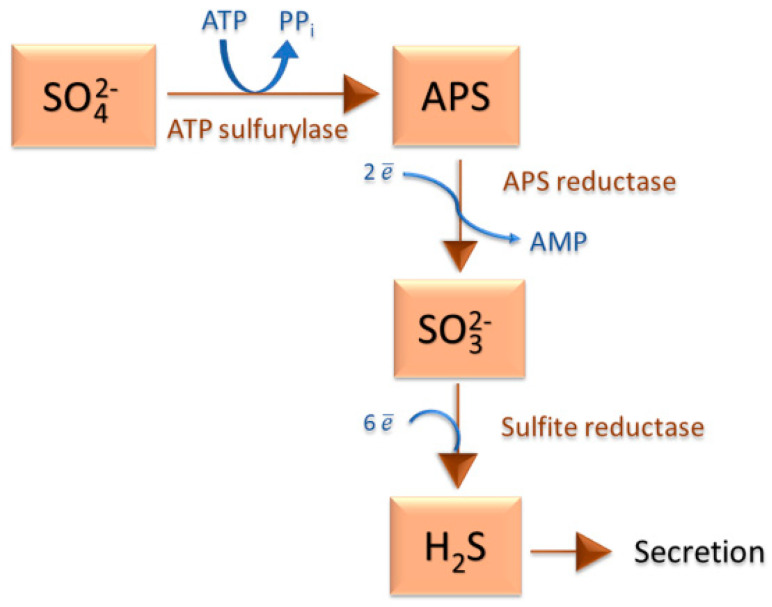
Scheme of the dissimilatory sulfate reduction.

**Figure 2 ijms-22-04007-f002:**
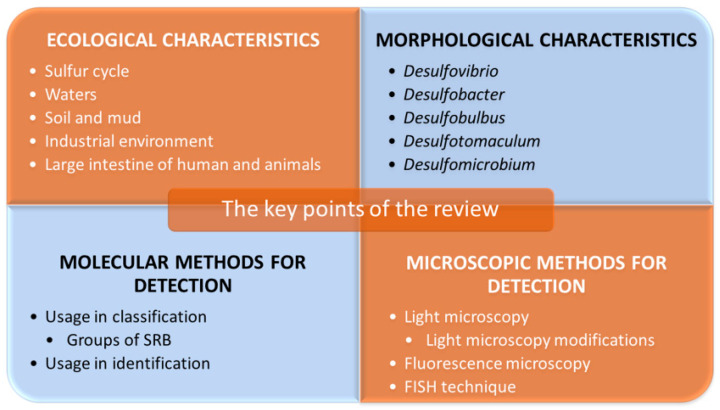
Main points of the review.

**Figure 3 ijms-22-04007-f003:**
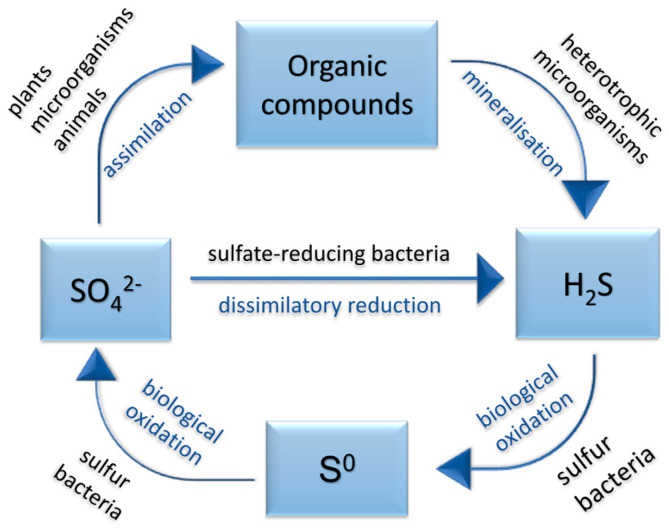
Scheme of the microbial sulfur cycle (modified from Tang et al. [29]).

**Figure 4 ijms-22-04007-f004:**
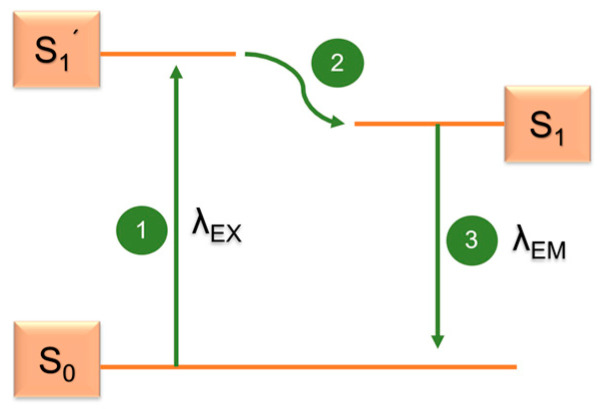
Jablonski diagram describing the phenomenon of fluorescence (modified from Johnson [110]).

**Figure 5 ijms-22-04007-f005:**
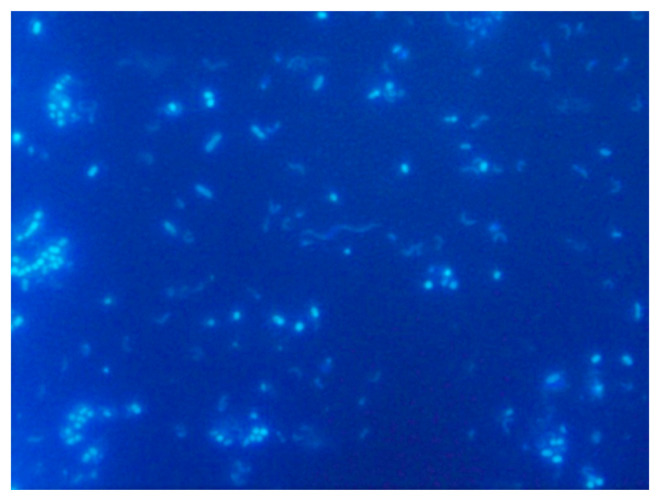
DAPI staining performed on a mouse gut sample. Visible cocci, rods, curved rods (possibly *Desulfovibrio*) and spiral-shaped cells (own micro-photograph).

**Figure 6 ijms-22-04007-f006:**
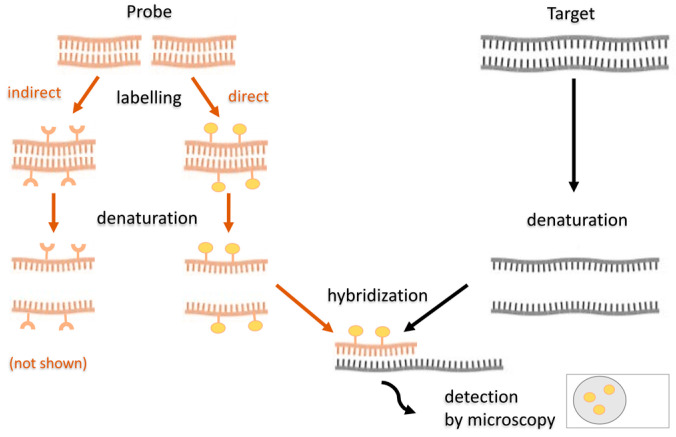
Scheme of the fluorescence in situ hybridization using a fluorescently (directly) labeled probe (modified from Speicher and Carter [116]).

**Figure 7 ijms-22-04007-f007:**
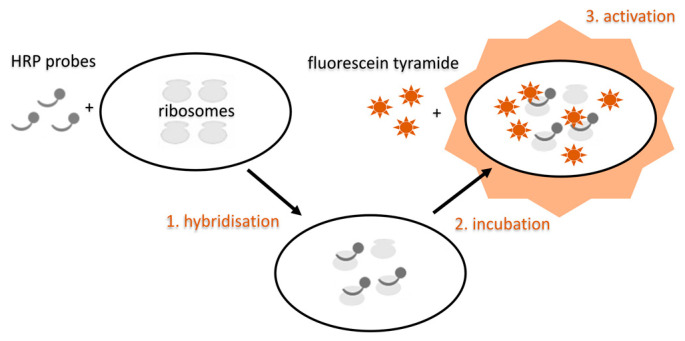
Scheme of tyramide signal amplification FISH in a bacterial cell. 1. hybridization with HRP-labelled probe. 2. incubation with fluorescein tyramide. 3. The activation of fluorescein tyramide and its binding to the cell’s electron-rich compartments, leading to fluorescence (modified from [128].

**Table 1 ijms-22-04007-t001:** Habitats of selected members of the genus *Desulfovibrio* (modified from Faque [10]).

Species of *Desulfovibrio*	Source	Habitat
*D. africanus*	NCIB 8401	well water
*D. alcoholovorans*	DSM 5433	fermenter
*D. carbinolicus*	DSM 3852	wastewater treatment plant
*D. desulfuricans*	DSM 642	soil
*D. piger*	ATCC 29098	human feces
*D. gigas*	NCIB 9332	pond water
*D. longus*	DSM 6739	oil field
*D. sulfodismutans*	DSM 3696	freshwater mud
*D. termitidis*	DSM 5308	intestines of termites

**Table 2 ijms-22-04007-t002:** Habitats of selected mesophilic sulfate-reducing bacteria (SRB) (modified from Faque [10]).

Species	Source	Habitat
*Desulfobulbus marinus*	DSM 2058	marine mud
*Desulfomicrobium apsheronum*	AUCCM 1105	oil field
*Desulfobotulus sapovorans*	DSM 2055	freshwater mud
*Desulfohalobium retbaense*	DSM 5692	hypersaline lake sediments

**Table 3 ijms-22-04007-t003:** Selected foods with high sulfate content, sorted in descending order (modified from Florin et al. [52]).

Food	Sulfate Content (µmol g^−1^)
Dried apples	49
Dried apricots	30
Commercial dark wheat bread	15
Commercial light wheat bread	13
Soy flour	12
Sausage	10
Almonds, hazelnuts	9
Brussels sprouts, broccoli	9
Cabbage (red, white)	8
Commercial rye bread	8
Jams, marmalades	7
Muesli	6

**Table 4 ijms-22-04007-t004:** Groups of SRB based on rRNA sequences with their representatives and important molecular and biochemical characteristics (data taken from Castro et al. and Daly et al. [81,82]).

Group	Genus	G+C %	Desulfoviridin	Cytochromes	Acetate Oxidation
Gram-positive sporulating					
1	*Desulfotomaculum*	48–52	–	*b*, *c*	complete/partial
Gram-negative mesophilic					
2	*Desulfobulbus*	59–60	–	*b*, *c*, *c*_3_	partial
3	*Desulfobacterium*	41–52	–	*b*, *c*	complete
4	*Desulfobacter*	44–46	–	no data	complete
5	*Desulfococcus*	46–57	+/–	*b*, *c*	complete
	*Desulfosarcina*	51	–	*b*, *c*	complete
6	*Desulfovibrio*	49–66	+/–	*c*_3_, *b*, *c*	partial
	*Desulfomicrobium*	52–67	–	*b*, *c*	partial
Thermophilic bacterial					
*Thermodesulfobacterium*		30–38	–	*c*_3_, *c*	partial
Thermophilic archaeal					
*Archaeoglobus*		41–46	–	no data	partial

**Table 5 ijms-22-04007-t005:** Overview of important fluorophores and their excitation (λ_EX_) and emission (λ_EM_) wavelengths.

Fluorophore	λ_EX_	λ_EM_
Fluorescein isothiocyanate (FITC)	494 nm	518 nm
Tetramethylrhodamine-isothiocyanate (TRITC)	555 nm	580 nm
Hoechst 33342 and 33258	352 nm	461 nm
Propidium iodide (PI)	536 nm	617 nm
Ethidium bromide	518 nm	605 nm
DAPI (4′,6-diamidino-2-phenylindole)	358 nm	461 nm
Cy3	550 nm	570 nm
Cy5	649 nm	670 nm

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
