# Peer review of "Microscopic Methods for Identification of Sulfate-Reducing Bacteria from Various Habitats"

_ijms, 2021, doi:10.3390/ijms22084007_

Round 1

Reviewer 1 Report

I was a little bewildered when I received a review on sulfate-reducing microorganisms for my consideration. The reason is that there are numerous reviews and books on this subject. In this respect any new review should be specifically coined to target some new or underreported aspects. The Authors seemed understood  that, and it has been reflected in the title. Yet, they turned wrong way started covering ecology, metabolism and some genera descriptions. That part ,in my opinion should be avoided, as they did it not on par to brilliant reviews by F. Widdel, F. Bak, JM Odom and such that have become reference books for long ago. The Authors failed contributing anything new in these aspects, just retelling a common knowledge. Their review would pan out much better if they focused on methods and applications in medical area. Even if the review meant for students or people who never dealt with sulfate-reducing microorganisms before, it would be much better to give a brief introduction with references on first-class reviews. Some terms , such as SRB, should be avoided, because professionals use sulfate reducing microorganism (SRM) or sulfate reducing prokaryotes (SRP) now days as sulfate reducers comprise both bacteria and archaea.

The description of SRP in the review was limited to five genera. Why five? There are at least 30 validated genera of sulfate-reducing deltaproteobacteria, plus sulfate reducing clostridia and archaea. The readers who are interested could be referred to Bergey’s Manual of Systematic Bacteriology when all genera are covered. Thus, I think the Authors should re-think their review focusing it on methods and its applications in medical field. To this end, do not forget metagenomics, which is widely used for characterization of complex consortia including gut microbiota. Metagenomics is used more often than FISH currently. A whole genome sequence is now a requirement for any new species description. In this way DNA-DNA hybridization is performed in silico mostly. I would avoid descriptions of a theoretical basis for methods that are now conventional.  English language requires slight improvement.

Author Response

Thanks. The response letter is attached.

Reviewer 2 Report

Dear authors, this is a well-intended manuscript but it needs to be more labored.

The title of the manuscript is in fact the section 5 of the same, but the whole manuscript is a compendium about diverse information regarding sulfate reducer bacteria. In some parts or sentences it seems like the authors talk about their own experimental work (their own observations) but I guess it is just the style used for writting and this is really a bibliographic review (or…where is the information regarding methods/experiments?). It remains confused.

I think the manuscript needs a great reestructuring: it should focus on the main objective about the usefulness of the microscopy techniques to identify SRB and why they are accurate. There is a lot of information about SRB but it is still a bit disconnected in some parts. The main section (considering the title of the manuscript) (5) is just a summary about fundaments and operation of  microscopy techniques but there is an absolute lack of relation with SRB (they are just mentioned in line 874). The last conclusions is good but the text in the manuscript does not support them.

I think it should be resubmitted in a different way.

Author Response

(The authors gave the same response as above.)

Round 2

Reviewer 1 Report

Thanks. I  checked the revised version of  a review article “Microscopic Method for Identification of Sulfate-Reducing Bacteria from Various Habitats”. 

First of all, the title does not reflect the content of the review, as the Authors are going far beyond microscopic methods of identification of SRB and other identification methods in general. They are trying to cover an enormous field of ecology and taxonomy of sulfate-reducing organisms as well as their industrial applications. They are not up to the task. Trying to produce a big picture they failed to do it on par with the current professional level and sometimes even made unforgivable mistakes that can confuse and mislead a reader.

I already suggested to the  Authors  to limit their review by methods for identification and focus on medical applications. Yet, they added the information and references I suggested but didn’t omit anything from the paper. As a result, it’s overloaded with unnecessary and sometimes erroneous information.  I do not have much time to review this paper as it is.  I think the Authors have to modify it adjusting to their own title.